Lipopolysaccharide-activated macrophages regulate the osteogenic differentiation of bone marrow mesenchymal stem cells through exosomes

Song Xiao 1 2
Xue Yiwen 1 2
Fan Siyu 1 2
Hao Jing 3
Deng Runzhi doctord@163.com 1
1 Department of Oral and Maxillofacial Surgery, Nanjing Stomatological Hospital, Medical School of Nanjing University , Nanjing , Jiangsu , China
2 Central Laboratory of Stomatology, Nanjing Stomatological Hospital, Medical School of Nanjing University , Nanjing , Jiangsu , China
3 Department of Orthodontics, Nanjing Stomatological Hospital, Medical School of Nanjing University , Nanjing , Jiangsu , China
Orsini Giovanna
Electronic publication date: 2022 May 13
Publication date: 2022
Volume: 10
Electronic Location ID: e13442
Received 2021 Dec 14; Accepted 2022 Apr 25
Copyright: ©2022 Song et al.
Copyright year: 2022
Copyright holder: Song et al.
License: This is an open access article distributed under the terms of the Creative Commons Attribution License, which permits unrestricted use, distribution, reproduction and adaptation in any medium and for any purpose provided that it is properly attributed. For attribution, the original author(s), title, publication source (PeerJ) and either DOI or URL of the article must be cited.
License URL: https://creativecommons.org/licenses/by/4.0/

Keywords: Exosome, Periodontitis, Macrophage, Stem cell, Osteogenesis differentiation

Funding: Project of Invigorating Health Care through Science, Technology and Education, Jiangsu Provincial Medical Youth Talent QNRC2016122 Nanjing Medical Science and Technique Development Foundation YKK19092 YKK21183 This research was supported by the Project of Invigorating Health Care through Science, Technology and Education, Jiangsu Provincial Medical Youth Talent under Grant [grant number QNRC2016122], the Nanjing Medical Science and Technique Development Foundation (YKK19092), (YKK21183). The funders had no role in study design, data collection and analysis, decision to publish, or preparation of the manuscript.

==============================
Background

Periodontal tissue regeneration is the ultimate goal of periodontitis treatment. Exosomes are nanoscale vesicles secreted by cells that participate in and regulate the physiological activities between cells. However, the relationship between inflammatory macrophage-derived exosomes and osteoblast differentiation in periodontitis has not been thoroughly reported. Here, we attempt to explore the role of inflammatory macrophage-derived exosomes in crosstalk with osteoblasts.

Methods

Porphyromonas gingivalis lipopolysaccharide was used to stimulate macrophages and inflate their inflammatory cellular state. Exosomes were extracted from inflammatory macrophages using supercentrifugation, and their characteristics were detected by transmission electron microscopy, particle size analysis, and Western blotting. Exosome uptake bybone marrow mesenchymal stem cells (BMSCs) was observed by fluorescence microscopy. The effects of exosomes on the BMSC inflammatory response and on osteogenic differentiation were detected by quantitative polymerase chain reaction and Western blot analysis. Alkaline phosphatase activity was tested for verification.

Results

We successfully extracted and identified inflammatory macrophage-derived exosomes and observed that BMSCs successfully took up exosomes. Inflammatory macrophage-derived exosomes upregulated the expression levels of the inflammatory factors interleukin-6 and tumour necrosis factor-alpha in BMSCs and mediated inflammatory stimulation. Additionally, they inhibited the transcription levels of the osteogenic genes alkaline phosphatase, type I collagen, and Runt-related transcription factor 2 as well as the alkaline phosphatase activity, while the use of the exosome inhibitor GW4869 attenuated this effect.

Conclusion

Our study shows that macrophages in periodontitis can mediate inflammatory stimulation and inhibit the osteogenic differentiation of bone marrow mesenchymal stem cells through the exosome pathway. Interference with exosome secretion is likely to be a promising method for bone tissue regeneration in inflammatory states.

Introduction

Chronic periodontitis is one of the most common diseases in the oral and maxillofacial regions. Gram-negative anaerobic bacteria, especially Porphyromonas gingivalis, are the main pathogenic bacteria of periodontitis. Chronic periodontitis is characterized by the loss of periodontal supporting tissue caused by bacterial infection and the host immune-inflammatory response (Cochran, 2008), leading to alveolar bone absorption and tooth loosening and shedding (Chen et al., 2020c). Reasonable control of periodontitis can help patients regain chewing function, improve quality of life and promote mental health. The core goal of clinical periodontal therapy is still to remove calculus, plaque, and lesions from root surfaces, eliminate local inflammation of periodontal tissue and prevent disease progression. However, the ultimate goal of periodontal therapy, namely the reconstruction of supporting tissue structure, cannot yet be achieved (Barnes et al., 2013).

Periodontal tissue engineering with stem cells as the core element is a promising future treatment of periodontitis. Alveolar bone regeneration is dependent on bone marrow mesenchymal stem cells (BMSCs), periodontal ligament stem cells and osteoblast differentiation. However, BMSCs are also affected by inflammatory processes. It has been reported that inflammatory or immune-related cytokines such as tumour necrosis factor-α (TNF-α), interferon-γ, and interleukin-17 (IL-17) affect BMSC differentiation and induce apoptosis (Han et al., 2017). Therefore, clarifying the effect of the inflammatory environment on BMSC osteogenic differentiation and exploring the specific mechanism behind it may help reduce the adverse impact of the inflammatory environment on BMSCs and promote their osteogenic differentiation, thus providing a potential treatment for periodontal tissue regeneration. When inflammation is confined to periodontal tissue, macrophages are the earliest immune cells recruited to the inflammatory site. A large number of macrophages can be seen in the periodontal tissue of patients with periodontitis, and a large number of proinflammatory factors and bone resorptive factors, such as interleukin 1β, interleukin 6 (IL-6), and TNF-α, can be detected in gingival crevicular fluid (Lam et al., 2014). Macrophages play an indispensable role in the occurrence and development of periodontitis. It has been shown that the coculture of macrophages and BMSCs can lead to the upregulation of the receptor activator of NF-κB ligand in an inflammatory environment, thereby affecting osteoclast formation and bone remodelling (Xiao et al., 2018). Macrophage paracrine signalling is an important pathway that regulates BMSC osteogenic differentiation. The uptake of paracrine vesicles, exosomes, and other cytokines by BMSCs can upregulate the transcription levels of alkaline phosphatase (ALP), Runt-related transcription factor 2 (RUNX2), osteocalcin, and osteopontin and the protein expression levels of RUNX2 and osteocalcin, enhancing the activity of ALP (Chen et al., 2021). Additionally, lipopolysaccharide-activated macrophages can promote the migration of BMSCs through paracrine IL-6 and inducible nitric oxide synthase (Lei et al., 2021). In periodontitis, infiltrating macrophages can regulate the function of BMSCs through paracrine cytokines. However, further studies are needed to determine whether exosomes mediate the regulation of BMSCs by macrophages.

Exosomes are nanoscale vesicles with a diameter of 30–140 nm that are secreted by cells and contain complex RNA and proteins (Yang et al., 2017). They function mainly as carriers of information between donor and recipient cells (Milane et al., 2015). Under normal circumstances, most cells, including immune cells, can secrete exosomes, such as lymphocytes, macrophages, and dendritic cells. Additionally, the secretion amount and contents of exosomes may change under the stimulation of pathogens (Chen et al., 2020b). It has been shown that in the microenvironment of hypoxia and serum deprivation, exosomes from M1-type macrophages can induce the apoptosis of BMSCs by delivering miRNA-222 to BMSCs (Qi et al., 2021). Additionally, M2-type macrophage-derived exosomal miRNA-5106 induces BMSCs to develop into osteoblasts by targeting salt-induced kinases 2 and 3 (Xiong et al., 2020). However, the effect of inflammatory macrophage-derived exosomes on BMSCs under the conditions of periodontitis has not been further reported. Therefore, exploring the role of macrophage-derived exosomes in regulating BMSC osteogenic differentiation in periodontitis is expected to provide a new solution for the realization of ideal periodontal tissue regeneration.

In this study, we successfully extracted, identified, and characterized inflammatory macrophage-derived exosomes activated by Porphyromonas gingivalis lipopolysaccharide (P.g-LPS). Additionally, we verified that macrophages could regulate BMSCs by secreting exosomes and inhibiting their osteogenic differentiation in the periodontitis environment, which may provide a potential treatment for periodontal tissue regeneration.

Materials and Methods

Cell culture and treatment

Mouse mononuclear RAW264.7 macrophages (purchased from Shanghai Cell Bank, Chinese Academy of Sciences) were cultured in Dulbecco’s modified Eagle’s medium (DMEM) (Gibco, Waltham, MA, USA) containing 10% foetal bovine serum (Gibco, Waltham, MA, USA) and 1% penicillin–streptomycin in a humid environment containing 5% CO2 at 37 °C. The cells were inoculated into 6-well plates at a density of 2 × 105 cells per well for 12 h and were then pre-treated with 100 ng/ml P.g-LPS (Sigma Aldrich, USA) or phosphate-buffered saline (PBS) (Gibco, Waltham, MA, USA) for 0 h, 1 h, 3 h, 6 h, 12 h, 24 h. The pre-treatment and control group samples were collected at different time points, using a quantitative polymerase chain reaction to analyse the expression differences of pro-inflammatory cytokines IL-6 and TNF-α. The time point with the highest expression of inflammatory genes was selected as the P.g-LPS stimulation time for the follow-up experiment. After 6 h of P.g-LPS stimulation, the cells were washed three times with PBS, then the medium was repalced without P.g-LPS, and the cells continued to culture for 24 h or 48 h. The cell culture supernatants were collected, then the enzyme-linked immunosorbent assay (ELISA) was used to determine the IL-6 and TNF-α protein levels in the supernatants and to verify that the extracellular inflammatory environment in macrophages persisted for more than 48 h after P.g-LPS stimulation.

Mouse bone marrow mesenchymal stem cells (purchased from Zhongqiaoxinzhou Biotech, Shanghai, China) were cultured in DMEM/F12 medium (Gibco, Waltham, MA, USA) containing 10% foetal bovine serum and 1% penicillin–streptomycin in a humid environment containing 5% CO2 at 37 °C. To evaluate the osteogenic differentiation of BMSCs, cells were incubated with corresponding CM supplemented with the ingredients of the osteogenic (OS) induction medium containing 0.1 µM dexamethasone, 10 mM β-glycerophosphate, and 50 µM ascorbic acid. BMSCs treated with the OS induction medium were then used for the Western blotting and Alkaline phosphatase (ALP) staining assays.

Real-time quantitative PCR

Total RNA was extracted from cells using the Cell Total RNA Extraction Kit (Novizan, Nanjing, China) and then further reverse transcribed to cDNA using the PrimeScript™ II 1st Strand cDNA Synthesis Kit for real-time polymerase chain reaction (Novizan, Nanjing, China). The qRT-PCR was carried out using ChamQ Universal SYBR qPCR Master Mix (Novizan, Nanjing, China). Every reaction was performed in a final volume of 10 µl containing 4 µl of cDNA, 0.5 mM of each primer, and 5 µl of ChamQ Universal SYBR qPCR Master Mix. The amplification was carried out as outlined in the instructions. The relative expression levels of the target genes, including IL-6, TNF-α, ALP, COL-I, and RUNX2 were then calculated by the comparative 2-ΔΔCt method using StepOne Software version 2.1.22. All the samples were run in triplicate and normalized to GAPDH. Primer sequences are shown in Table 1.

Table 1 Sequences of primers used in quantitative real-time PCR (qRT-PCR) analysis.

Gene	Primer sequence	
	Forward	Reverse	
TNF-α	GGAGGGGTCTTCCAGCTGGAGA	CAATGATCCCAAAGTAGACCTGC	
IL-6	CTTGGGACTGATGCTGGTGACA	GCCTCCGACTTGTGAAGTGGTA	
ALP	ACGGCGTCCATGAGCAGAACTA	CAGGCACAGTGGTCAAGGTTGG	
COL-1	GAGTCAGCAGATTGAGAACATCC	AGTCAGAGTGGCACATCTTGAG	
RUNX2	CCCAGGCAGTTCCCAAGCATTT	GGTAGTGAGTGGTGGCGGACAT	
GAPDH	AGGTCGGTGTGAACGGATTTG	GGGGTCGTTGATGGCAACA	

Enzyme-linked immunosorbent assay (ELISA)

After 6 h of P.g-LPS stimulation, the cells were washed three times with PBS, then the medium was replaced without P.g-LPS, and the cells continued to culture for 24 h or 48 h. The cell supernatant was then collected, and soluble protein concentrations of IL-6 and TNF-α in the supernatant were detected using ELISA kits (Enzyme Immunoassay, Wuhan, China) according to the instructions. The determination ranges were as follows: TNF-α, 25–800 pg/mL; IL-6, 3–120 pg/ml. Samples were evaluated in triplicate.

Conditioned culture medium collection

The cells were inoculated at a density of 2 × 107 cells per well in a 150 mm dish overnight and then stimulated with 100 ng/mL P.g-LPS for 6 h. After three washes with PBS, the cells were placed in DMEM containing 10% foetal bovine serum which hads been centrifuged at 11,000× g for 20 h to eliminate exosomes and 1% penicillin–streptomycin with or without 10 µM GW4869 (MCE, Princeton, New Jersey, USA) and further cultured for 24 h. The cell culture supernatants were collected. Cells and cell debris in the supernatant were removed by centrifugation at 4 °C at 1,500× g for 20 min, and particles with a particle size greater than 200 nm were removed by filtration with a 0.22-µm filter. Conditioned medium (CM) with or without GW4869 was labelled CM+GW4869 and CM, respectively.

Extraction, quantification, and identification of exosomes

The collected supernatant CM was centrifuged with a high-speed centrifuge (Thermo, Waltham, MA, USA) at 4 °C for 17,000× g for 15 min, and then residual organelles were removed from the supernatant. A superspeed centrifuge (Thermo, Waltham, MA, USA) was then used at 4 °C and 110,000× g for 80 min. After discarding the supernatant, the trace white precipitate at the bottom of the centrifuge tube was considered as exosomes, which were resuspended in PBS.

Ten-microlitre exosome samples were subjected to lysis on ice for 10 min with the same amount of RIPA lysate, followed by the protein quantification of exosomes according to the instructions of the BCA kit (Beyotime, Shanghai, China). Then, 10 µl of exosomes was dropped onto the copper wire for precipitation for 1 min, and the float was absorbed by filter paper. Then, 10 µl of 2% uranium acetate was dropped onto the copper wire for staining for 1 min. After natural drying at room temperature, exosomes were characterized by a transmission electron microscope (Hitachi, Chiyoda, Japan) at 100 kV. Dynamic light scattering was performed using a particle size analyser (NanoFCM, Xiamen, China) to measure the size distribution of the exosomes. Western blotting, with 12 µl samples of protein per well, was used to detect the expression of the exosome surface-enriched proteins CD9, CD81, and TSG101. Calnexin was used as the internal reference.

Labelling of exosomes and incubation of recipient BMSCs

The quantitative exosome suspension was labelled with 6 µl of fluorescent labelling probe PKH67 (Merck, Darmstadt, Germany) and incubated at 37 °C for 5 min. Then, the mixture was centrifuged at 190,000× g for 2 h at 4 °C. The supernatant was discarded and suspended again with PBS to remove the excess dye. BMSCs were inoculated at 5 × 105 cells per well in 6-well plates. The labelled exosomes with a final concentration of 50 µg/mL were incubated with BMSCs for 24 h. Fifty microlitres of Hoechst solution was added to each well, and the plates were placed at room temperature for 30 min under dark conditions. The uptake of exosomes was observed under a 600x fluorescence confocal microscope (FV3000; Olympus, Tokyo, Japan).

Exosomes and conditioned medium transfection

The experiment was divided into four groups: ordinary exosome group, inflammatory exosome group, CM group, and CM+GW4869 group. BMSCs were inoculated in 6-well plates at a density of 2 × 105 cells per well for 12 h. Then, for the inflammatory exosome group, cells were treated with 2.5 ml exosome-free medium containing 50 µg/ml exosomes secreted by macrophages stimulated by P.g-LPS. The CM and CM+GW4869 groups were treated with 2.5 mL of undiluted CM or CM+GW4869. The ordinary exosome group was treated with 2.5 ml of exosome-free medium containing 50 µg/ml exosomes secreted by macrophages under normal conditions. Forty-eight hours after transfection, cells were collected for subsequent experiments.

Western blotting

Proteins in exosomes or cells were lysed and extracted using a whole-protein extraction kit (KeyGEN, Nanjing, China). Then, the proteins were separated by SDS–PAGE and transferred to NC membranes, blocked in 5% skim milk at room temperature for 1.5 h, and incubated with the following primary antibodies: anti-CD9 (1:300; Abcam, Cambridge, MA, USA), anti-CD63 (1:300; Abcam, Cambridge, MA, USA), anti-TSG101 (1:300; Abcam, Cambridge, MA, USA), anti-calnexin (1:300, Abcam, Cambridge, MA, USA), anti-TNFα (1:1,000; Abcam, Cambridge, MA, USA), anti-IL-6 (1:1000; Abcam, Cambridge, MA, USA), anti-ALP (1:1,000, Boaosen, China), anti-RUNX2 (1:1,000; Abcam, Cambridge, MA, USA), anti-COL-1 (1:1,000; Abcam, Cambridge, MA, USA), and anti-GAPDH (1:5,000; Abcam, Cambridge, MA, USA). The blot was then stained with horseradish peroxidase (HRP)-conjugated goat anti-rabbit secondary antibody (KGAA35; KeyGEN, Nanjing, China). Finally, the protein expression level was detected using a chemiluminescence detection system. Each experiment was repeated 3 times.

Determination of ALP and ALP staining

BMSCs were plated in 6-well plates at a density of 2 × 105 cells per well for 12 h and incubated for 7 days and 14 days under the same treatment conditions as above, and the solution was changed every 3 days. After 7 days, the cells were collected and lysed, and the total protein amount of each group was determined using the BCA kit (Beyotime, Shanghai, China). After verifying that the total protein amount of the ordinary exosome group, the inflammatory exosome group, the CM group, and the CM+GW4869 group were all at the same level, the alkaline phosphatase activity of each group was detected using the ALP kit (Built, Nanjing, China). Each experiment was repeated 3 times.

After 14 days, a BCIP/NBT alkaline phosphatase color development kit (Beyotime, Shanghai, China) was used according to the provided directions. The cells were washed three times with PBS and fixed with 4% paraformaldehyde for 30 min, then treated with BCIP/NBT substrate for 10 h, A microscope was used to analyse the colorimetric changes and a scanner was used to image the stained cells. Absorbance was then measured at 450 nm. Experiments were repeated in triplicate.

Data analysis

All numerical data are presented as the mean and standard deviation. Comparisons between two groups were conducted using independent samples t-tests. One-way ANOVA was used to analyse differences among more than two groups followed by Tukey’s post-hoc test when data were normally distributed and group variances were equal. All statistical analyses were performed using the GraphPad Prism 8 (GraphPad Software, Inc., La Jolla, CA, USA). P < 0.05 was considered significant.

Results

Release of exosomes from RAW264.7 cells under P.g-LPS stimulation

P. gingivalis LPS-stimulated (100 ng/mL) RAW264.7 cells produced an inflammatory response. The qPCR results revealed that the gene expression levels of inflammatory factors TNF-α and IL-6 increased with time with the expression levels of these inflammatory factors reaching their highest level at about 6 h. (Figs. 1A–1B). RAW264.7 cells were pretreated for 6 h for subsequent experiments. ELISA results showed that compared with the control group, the soluble protein TNF-α and IL-6 contents in the 24 and 48 h groups were higher than those in the control group (Figs. 1C–1D), suggesting that the inflammatory state of RAW264.7 cells lasted for at least 48 h after 6 h of pretreatment with P. gingivalis LPS.

Figure 1 Macrophage secreted exosomes after P.g-LPS stimulation.

(A–B) The mRNA expression of proinflammatory mediators IL-6 and TNF-α of macrophages after stimulation with 100 ng/mL P.g-LPS for 1–24 h. (C–D) The expression of soluble proteins IL-6 and TNF-α in macrophage extracellular environment at 24 h and 48 h after stimulation by 100 ng/mL P.g-LPS for 6 h. (E) Microscopic structure of exosomes (scale bar: 1 µm, 500 nm, 200 nm, 100 nm) in the transmission electron microscope. (F) Exosome particle size distribution. (G) Exosome concentration determination diagram. (H) The expression of exosome surface enriched proteins CD81, CD9, TSG101 and negative marker protein Calnexin, the control group was the RAW264.7 cell lysate. An asterisk (*) indicates P < 0.05, two asterisks (**) indicate P < 0.01, three asterisks (***) indicate P < 0.001, four asterisks (****) indicate P < 0.0001, ns indicates P > 0.05, ns indicates no significant difference. All * are compared with the control group.

After 325 ml samples of supernatant from RAW264.7 exosomes were extracted using the ultracentrifugation ultrafiltration method, the samples were suspended in 200 µl PBS. The protein concentration was approximately 2.23 mg/mL by BCA protein determination, and the collected RAW264.7 exosomes were approximately 446 µg. Subsequently, the exosomes were characterized and identified by transmission electron microscopy, particle size analysis, and Western blotting. Under transmission electron microscopy, exosomes were observed as vesicles with a double-layer membrane structure outside and low electron density materials with uneven density inside (Fig. 1E). Most exosomes have particle sizes ranging from 70 to 100 nm, with an average particle size of 81.20 nm (Figs. 1F–1G). Compared with purified exosomes, the exosomal surface-enriched proteins CD9 and TSG101 were expressed in the extracted exosomes with high abundance, while the negative marker protein calnexin was not expressed (Fig. 1H). These results indicated that exosomes from RAW264.7 cells were successfully extracted.

Effect of inflammatory macrophage-derived exosomes on BMSC osteogenic differentiation

To verify whether inflammatory macrophage-derived exosomes mediate the communication between BMSCs and the regulation of osteogenic differentiation in BMSCs, we incubated BMSCs with inflammatory macrophage-derived exosomes for 24 h, and confocal microscopy results showed that BMSCs successfully absorbed exosomes into the cytoplasm around their nuclei (Fig. 2A), which is a prerequisite for exosome-mediated communication.

Figure 2 Exosome uptake by BMSCs and exosomes mediating the inflammatory response.

(A) Images of exosome uptake by BMSCs in fluorescence confocal microscopy. BMSCs nuclei were stained with Hoechst (blue), BMSCs Cytoskeleton were stained with FITC-Phalloidin (red), macrophage-derived exosomes were labelled with PKH67 (green). (B–D) The protein expression of TNF-α and IL-6. (E–F) The mRNA expression levels of TNF-α and IL-6. An asterisk (*) indicates P < 0.05, two asterisk (**) indicates P < 0.01, three asterisks (***) indicate P < 0.001, four asterisks (****) indicate P < 0.0001, ns indicate P > 0.05, ns indicates no significant difference.

Subsequently, we examined the changes in the gene and protein expression levels ofinflammatory factors TNF-α and IL-6 to determine whether exosomes can mediate the inflammatory state in recipient cells. As shown in Figs. 2B–2F, BMSCs were treated with ordinary exosomes, inflammatory exosomes, or CM. Compared with the ordinary exosome group, TNF-α, IL-6 gene, and protein expression levels were significantly upregulated in the inflammatory exosome group. When GW4869 inhibited exosome release, compared with the CM group, the expression levels of the TNF-α and IL-6 genes in the CM+GW4869 group decreased. However, there was no significant difference in TNF-α and IL-6 protein levels between the two groups. These results suggest that exosomes are involved in the mediating process of BMSC inflammatory stimulation.

The effects of ordinary exosomes, inflammatory exosomes, and CM on BMSC osteogenic differentiation are summarized in Fig. 3. Compared with ordinary exosomes, inflammatory exosomes had a reduced expression of ALP, RUNX2, and collagen type I (COL-1) at both the gene and protein levels. Additionally, the expression levels of osteogenic genes ALP, RUNX2, and COL-1 were increased in BMSCs compared with CM after GW4869 treatment (Figs. 3A–3G). The alkaline phosphatase activity test and staining results were consistent with BMSC gene expression (Figs. 3H–3K).

Figure 3 Effects of inflammatory macrophage-derived exosomes on osteogenic indices and ALP activity of BMSCs.

(A) ALP mRNA expression, (B) COL-1 mRNA expression, (C) RUNX2 mRNA expression in BMSCs after transfection of ordinary exosomes, inflammatory exosomes, CM, and CM+GW4869 for 48 h. (D–G). The protein expression after BMSC transfection with ordinary exosomes, inflammatory exosomes, CM, and CM+GW4869 for 7 days. (H) BCA quantification of ALP activity. (I) ALP activity. (J) ALP staining images. (K) Quantitative mineralization assay of panel J. * indicates P < 0.05, two asterisk (**) indicate P < 0.01, three asterisks (***) indicate P < 0.001, four asterisks (****) indicate P < 0.0001, ns indicates no significant difference.

Discussion

Periodontitis disrupts the balance between bone formation and bone resorption in the alveolar bone, which increases bone loss in the alveolar bone (Xu et al., 2018). As an essential part of innate tissue immunity, the number of macrophages in the periodontal tissues of patients with chronic periodontitis is much higher than that in normal periodontal tissues. As the progenitors of osteoclasts, macrophages can induce bone resorption by secreting proinflammatory cytokines, and excessive cytokines can inhibit the differentiation, proliferation, and mineralization of osteoblasts. The degradation of the bone matrix provides attachment sites for osteoclasts, leading to further loss of alveolar bone tissue (He et al., 2018). Macrophages play a significant role in alveolar bone loss caused by chronic inflammation (Lam et al., 2016). Macrophages influence and regulate the reconstruction and loss of alveolar bone through paracrine signalling, but intercellular communication depends on not only cytokines and other extracellular vesicles but also exosomes. In this study, BMSC uptake of macrophage-derived exosomes was successfully observed through staining and confocal microscopy. As carriers of transmitted information, exosomes can effectively protect their contents from the influence of the extracellular environment in order to achieve directional delivery to recipient cells (Fais et al., 2013). Studies have shown that exosomes are involved in many inflammatory processes, such as acute lung injury, inflammatory bowel disease, and asthma inflammation. (Canas et al., 2021; Monsel et al., 2016; Zhang et al., 2019). We therefore hypothesized that exosomes play a role in the crosstalk between macrophages and BMSCs under inflammatory stimulation.

The composition of macrophage exosome proteins is mainly divided into two categories. One category includes the standard proteins that are ubiquitous in the formation and secretion of vesicles. These proteins include the transmembrane transport proteins, fusion-related proteins (such as Rab and GTPases), heat shock proteins (such as HSP70 and HSP90), tetrapeptide transmembrane proteins (such as CD63, CD81, and CD9), and ESCRT complex-related proteins (such as Tsg101 and Alix). Among these proteins, CD63, CD81, CD9, and TSG101 are highly enriched in exosomes and have become commonly used marker proteins of exosomes (Pegtel & Gould, 2019). The other category includes specific components closely related to macrophages. Compared with synthetic vectors such as liposomes and nanoparticles, exosomes have extensive and unique advantages in disease diagnosis and treatment due to their endogeneity and heterogeneity. However, the impurity and low yield of exosomes limit their clinical application as well as their application in scientific research. The commonly used techniques for exosome separation include ultracentrifugation, ultrafiltration, polymer precipitation, and immunoaffinity chromatography with different techniques being used for different purposes and applications. The most widely used separation technology is supercentrifugation, which is also currently considered the gold standard for exosome extraction and separation (Zhang et al., 2020). In this study, ultrafast centrifugation was used to extract macrophage-derived exosomes, and the marker proteins CD9, CD81, and TSG101 were identified. The International Society for Extracellular Vesicles (ISEV) pointed out that the identification of two characteristic proteins can prove that the extracted substances are exosomes (Zhang et al., 2020), and the reason why CD81 is not expressed in our extracted exosomes is that, although the positive indicator is the exosome marker protein, which is widely reported in many studies, some indicators may have no obvious bands due to low protein concentration or antibody immune typing. Therefore, our experimental results can also prove that we successfully extracted exosomes derived from macrophages, but our low extraction concentration and large sample size were still limitations of this study. The transmission electron microscopy results (Figs. 1E–1F) showed that the average particle size of macrophage-derived exosomes extracted in this study was 81.20 nm. Compared to nanomaterials of various sizes, exosomes of this size can effectively cross barriers (such as the plasma membrane and the blood/brain barrier), with lower metabolic efficiency and a longer-acting time. At the same time, exosomes have great potential to become drug delivery vectors suitable for delivering various small molecule drugs, proteins, nucleic acids, and gene therapy agents due to their natural material transport properties, inherent long-term recycling capacity, and excellent biocompatibility (Cho et al., 2018; Ohno et al., 2013; Pascucci et al., 2014). Based on confocal microscopy results, we observed an increased fluorescence intensity after adding macrophage-derived exosomes compared to the control group, indicating the successful internalization of exosomes and distribution of cells in the cytoplasm. The main internalization pathway of exosomes is through clathrin-independent endocytosis and pinocytosis (Costa Verdera et al., 2017), which involve endosomal acidification and membrane fusion between exosomes and target cells (Bonsergent et al., 2021). The efficiency of exosome uptake depends on cholesterol and tyrosine kinase activity as well as PH, temperature, and freezing and thawing times during the experiment (Cheng et al., 2019). The exosomes extracted in this study were stored at −80 °C, and follow-up experiments were conducted within 10 days to ensure the activity of exosomes and the uptake efficiency of target cells.

Gram-negative anaerobic bacteria, especially Porphyromonas gingivalis, are the main pathogenic bacteria of periodontitis, and their main pathogenic component, lipopolysaccharide, can induce bone destruction by triggering the release of proinflammatory mediators such as IL-6 and TNF-α. Stimulated by LPS, macrophages contribute to the development of inflammation by secreting many biologically active molecules, including proinflammatory cytokines and proteolytic enzymes. Additionally, studies have pointed out that there are significant differences in the contents of exosomes released by LPS-activated macrophages, indicating that exosomes released under different conditions may be involved in various functional and pathological processes (Raeven, Zipperle & Drechsler, 2018). LPS exosomes have been proven to contain a large number of proinflammatory factors. Among the protein components transmitted by exosomes, TNF, CCL3, CD40, and Serpine1 are closely related to inflammatory responses, and LPS exosomes are involved in the activation of various inflammatory signalling pathways. These pathways include the Nod-like receptor, IL-17, TNF, and Toll-like receptor signalling pathways (Wang et al., 2019). Previous studies have shown that 100 ng/mL LPS is sufficient to induce inflammatory responses in macrophages (Bode, Ehlting & Haussinger, 2012), which was consistent with our experimental results. After LPS treatment, the gene and protein expression levels of IL-6 and TNF-α in macrophages were increased, and this inflammatory state persisted after the removal of the stimulus. Additionally, our study also confirmed that inflammatory exosomes are involved in the inflammatory mediations of macrophages to BMSCs, which is consistent with previous studies (Wang et al., 2017).

Recent studies have found that bone formation, resorption, and remodelling are inseparable from the regulation of the immune system, and the communication between macrophages and other osteocytes plays an essential role in bone tissue homeostasis and new bone formation. Previous studies believed that macrophage products such as TNF-α, IL-6, IL-1β, etc., were the main immune factors involved in bone destruction and that they reduced the osteogenic differentiation potential of stem cells. At the same time, the secretion and delivery of exosomes was thought to be a wasteful mechanism, but there is growing evidence that exosomes are ubiquitous mediators of cellular communication in all cell types. In this study, we proved that without the participation of the above-mentioned inflammatory factors, exosomes also greatly affected the osteogenic differentiation of BMSCs, which is consistent with the research reported by Zhu et al. (2019). Still, some studies have concluded that in the early stages of inflammation, M1-type macrophage exosomes were able to promote BMSC osteogenic differentiation, which may be due to the lower concentration of exosomes used in this study. Interestingly, we found that the expression levels of inflammatory markers IL-6 and TNF-α decreased in the CM group compared with the ordinary exosome group, while the expression levels of the osteogenic markers ALP, RUNX2, and COL-1 increased. Therefore, we speculated that macrophage-derived exosomes mainly mediate inflammation and inhibit osteogenic differentiation during the crosstalk between macrophages activated by lipopolysaccharide and BMSCs. However, other substances that exist in the macrophage extracellular environment that can inhibit inflammation and promote bone differentiation were not found in this study. Other studies have confirmed that all macrophage subtypes, including M0, M1, and M2 macrophages, can promote the osteogenic differentiation of BMSCs, and M1 macrophages have the most significant influence on bone formation (Chen et al., 2020a), which is consistent with our observations in this study. Cells in the macrophage extracellular environment and their factor-specific roles require further research, which will be the goal of our future studies.

In this study, our experimental results clarify that macrophage-derived exosomes are involved in inhibiting the osteogenic differentiation of stem cells in inflammatory states, and that the use of exosome inhibitors can attenuate this effect. But as we mentioned before, there are multiple ways of communication and crosstalk between cells, and whether blocking exosome secretion can be used to restore the osteogenic function of stem cells in the inflammatory state needs further verification. In addition, based on the similar size of exosomes and nanomaterials and the advantages exosomes have of lower immunogenicity, higher biocompatibility, natural targeting ability, and good biological barrier permeability, their carrier functions are used for more complex scenarios than nanomaterials. However, it is necessary to understand the specific mechanism of exosomes for intercellular communication under physiological and pathological conditions. In follow-up experiments, we will further improve the specific mechanism of the exosome pathway to inhibit the osteogenic differentiation of stem cells, such as the main components and mode of action of inflammatory exosomes to support this study.

Conclusion

In conclusion, our results suggest that P.g-LPS-activated macrophage-derived exosomes can inhibit BMSC osteogenic differentiation by mediating inflammatory stimulation. At present, our study is expected to provide a potential treatment plan to improve the treatment effect in the application scenario of in vitro stem cell therapy and to improve the reconstruction of alveolar bone in periodontitis by regulating exosomes in the periodontal microenvironment.

Supplemental Information

Supplemental Information 1 Raw data for Figures 1A, 1B

Click here for additional data file.

Supplemental Information 2 Raw data for Figures 1C, 1D

Click here for additional data file.

Supplemental Information 3 Raw data for Figure 1G

Click here for additional data file.

Supplemental Information 4 Raw data for the BCA quantification

Click here for additional data file.

Supplemental Information 5 Raw data for Figures 2E, 2F, 3A, 3B, 3C

Click here for additional data file.

Supplemental Information 6 Raw data for Figures 3H, 3I

Click here for additional data file.

Supplemental Information 7 Images of western blots

Click here for additional data file.

Additional Information and Declarations

Competing Interests

Author Contributions

Data Availability

The authors declare there are no competing interests.

Xiao Song conceived and designed the experiments, performed the experiments, analyzed the data, prepared figures and/or tables, authored or reviewed drafts of the paper, and approved the final draft.

Yiwen Xue performed the experiments, analyzed the data, prepared figures and/or tables, and approved the final draft.

Siyu Fan performed the experiments, prepared figures and/or tables, and approved the final draft.

Jing Hao performed the experiments, authored or reviewed drafts of the paper, and approved the final draft.

Runzhi Deng conceived and designed the experiments, authored or reviewed drafts of the paper, and approved the final draft.

The following information was supplied regarding data availability:

The raw measurements are available in the Supplementary Files.

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
