# Peer review of "Lipopolysaccharide-activated macrophages regulate the osteogenic differentiation of bone marrow mesenchymal stem cells through exosomes"

_PeerJ, doi:10.7717/peerj.13442_

## Round 0.1 · original submission · Major Revisions

Serious concerns have been raised, please provide accurate revision of your manuscript. In particular, respond to the comment regarding the missing look at the biological functions of BMSCs.

·

Basic reporting

The manuscript entitled “Lipopolysaccharide-activated macrophages regulate osteogenic differentiation of bone marrow mesenchymal stem cells through exosomes” by Xiao Song et al. investigated that how inflammatory macrophage-derived exosomes affected osteogenic differentiation. Exosomes was extracted from lipopolysaccharide (LPS) -treated macrophages and characterized using transmission electron microscopy, particle size analysis, and Western blotting. Exosome uptake was verified by bone marrow mesenchymal stem cells (BMSCs) using fluorescence microscopy. They have found that LPS-treated macrophage-derived exosomes increased the mRNA and protein levels of the inflammatory factors, such as interleukin-6 and tumour necrosis factor-alpha in BMSCs. Inflammatory exosomes also suppressed osteogenic genes, including alkaline phosphatase, type I collagen, and Runt-related transcription factor 2 as well as the alkaline phosphatase activity. They concluded that inhibition of exosome secretion is a potential therapy to suppress bone tissue destruction during inflammation.

Experimental design

1. Key experiments should have three biological replicates (n=3). It was not described in the manuscript.
2. In Methods, statistical analysis should be elaborated. It is too simply just stating “One-way ANOVA was used to analyze differences among more than 2 groups”. One-way ANOVA is used to test whether it is statistically different among all compared groups. Multiple comparison is followed to test which two groups are different.

Validity of the findings

1. In Figure 2C-F, authors did not describe what comparisons the asterisks annotated. While authors stated the level of TNF-α protein remained stable, it showed significant (**) in Figure 2C.
2. Figure 2D showed that the protein level of IL-6 was significantly increased by inflammatory exosome. However, authors stated that “there was no significant difference in IL-6 protein levels between the two groups”.
3. Authors measured mRNA and protein expression of osteogenic markers using RT-PCR and Western blot. However, there are no phenotypic experiments to support the effect of inflammatory exosomes on osteogenic differentiation of BMSCs. They could perform direct staining of matrix components.
4. Why did authors use mouse mononuclear RAW264.7 macrophage as the model? The most relevant model is human macrophage isolated from healthy subjects and periodontitis patients.
5. The key experiments should be reproduced and validated in a second cell line.

Additional comments

The manuscript will benefit from English editing.

Reviewer 2 ·

Basic reporting

.

Experimental design

.

Validity of the findings

.

Additional comments

Review of article: "Lipopolysaccharide-activated macrophages regulate osteogenic differentiation of bone marrow mesenchymal stem cells through exosomes”
In this manuscript by Song et al, the authors examine the effects of exosomes secreted from inflammatory macrophages on the osteogenic differentiation of bone marrow mesenchymal stem cells (BMSC). The authors showed that BMSC uptake exosomes and thereby leading to increased expression of proinflammatory cytokines such as IL-6 and TNF-alpha. Moreover, authors also found that exosomes are responsible for inhibiting the osteogenic differentiation of BMSC as confirmed by downregulation of expression of several genes (alkaline phosphatase, type 1 collagen and RUNX2) involved in osteogenic differentiation of BMSC.
The manuscript could be strengthened by addressing the below listed points.
The main concerns regarding the manuscript are the experimental design, the lack of some important controls and the methods are not properly described in detail which creates great difficulty for the readers to understand. Several important references are missing throughout the manuscript and there are several places in manuscript where the sentences are unclear or incomplete. There are numerous minor errors in English expression and punctuation, and general typographic errors, throughout. I suggest the authors should check the entire manuscript carefully and make corrections.
Method section was written very vaguely and not well explained and hence need to improve to clarify for the reader.
Line 119 pretreated time point?
Line 121-126: Authors need to clarify for the readers that what method was used to analyze expression of IL-6 and TNF-alpha at different time points and moreover cell lysate or cell supernatant was used to analyze expression of genes? What do authors mean by remaining cells is it the cells left after the supernatant collection? And they were further incubated for another 24h or 48h? what is the stimulation time for LPS is unclear.
Line 138: Authors need to specify what was the method used to quantify mRNA or cite the reference of article for the method being used.
Line 141: Authors need to clearly specify the time of LPS stimulation since it is very confusing for the readers as it says 24h or 48h stimulation or did the author mean time of supernatant harvest after 6h LPS stimulation since according to figure legends it is not consistent.
Line 147: Specify the name of cell. What was the composition of exosome clearance medium? Or cite reference article.
Line 169: Authors need to specify how much amount of lysate was used to quantify protein expression by western blot? Also, need to mention that Calnexin was used as a negative marker for exosomes.
Line173: name of the fluorescent probe is missing
Line178: what time point Hoechst stain was added?
Line 180: Authors need to mention the name of the microscope, magnification lens used and describe in detail the method used to analyze the images
Line 188-190: Need to describe the method briefly to clarify readers and What reagent was used for transfection?
Line 200: Authors need to specify the secondary antibodies used.
Line 209: Authors need to specify what groups or refer to method section where already written.
Line 221-222: Authors need to readdress the results (Fig 1A-B) as IL-6 levels were still elevated at both 12h and 24h which is very much comparable to 3h. Any explanations since both cytokines show completely different pattern of gene expression at different time points. TNF-alpha has highest expression levels at 3h when compared to IL-6 which is highly expressed at 6h. Authors need to give strong evidence why they choose 6h timepoint for LPS stimulation in most experiments.
Line 226-227: How can the authors reach this conclusion from Fig 1C-D if the experiment was not performed beyond 48h timepoint. To demonstrate this author, need to perform ELISA at 72h timepoint and confirm whether the inflammatory state truly lasted only up to 48h.
Authors should also harvest cell culture supernatant samples at similar timepoints 1h,3h,6h,12h and 24h for performing ELISA. Any explanations why chosen different timepoints for mRNA and ELISA analysis (Fig1).
In Fig1H authors need to explain the control in western blot in figure legend.
Authors need to discuss this result why CD81 shows no expression on the exosomes when compared to control and CD9 showed lower expression levels when compared to control?
Figure legends are written very ambiguously and not well explained and hence need to improve to clarify for the reader. Correct ‘ns means non-significant’
Fig1D replace H by h in figure labelling
In Fig2C-D keep the figure labelling consistent in all graphs like in Fig2E-F. Similarly change in Fig 3.
I strongly recommend seeing the effect of exosomes when BMSC and macrophages are co-cultured in vitro since it will strengthen the significance and conclusions.
Additionally, authors should perform the experiment using TLR signaling blockers/inhibitors and observe the effects of exosomes on BMSC.
Authors need to discuss in detail the limitations of the study and what do they interpret from their results.

Annotated reviews are not available for download in order to protect the identity of reviewers who chose to remain anonymous.

·

Basic reporting

This manuscript aims to understand how inflammatory macrophage-derived exosomes regulate osteogenic differentiation of bone marrow mesenchymal stem cells. To achieve this, the authors utilized P. gingivalis LPS to stimulate mouse macrophages, then they used ultrafast centrifugation to extract macrophage-derived exosomes. After characterizing the exosomes, the authors further looked at the effects of exosomes on the BMSC inflammatory response and osteogenic differentiation by qPCR and Western blot, and they found that exosomes upregulated the expression levels of two inflammatory factors and decreased the expression of several osteogenic markers in BMSCs. At last, the authors demonstrated that exosome inhibitor GW4869 could attenuate the effect. While the results generally support the conclusion, the manuscript lacks more solid data, and the data analysis in this manuscript is flawed.

Experimental design

1 A major negative aspect of the work is that it does not have supportive data looking at the fundamental biological functions of BMSCs upon the uptake of inflammatory exosomes. Though the authors thoroughly checked the expression of osteogenically marker genes as well as ALP activity, and found statistical differences, this does not necessary mean that there would be biological differences. Especially by comparing the data from cells treated with CM to those from cells treated with CM+GW4869, the differences of protein expression were minor. It would strengthen this study a lot if the authors could also look at the biological functions of BMSCs, such as their migration and mineralization, et al.
2 Please indicate in line 149 what concentration of GW4869 was used.

Validity of the findings

1 The most important issue is the statistical analysis:
a. If one-way ANOVA is used, the authors should clearly point out the difference between each pair of samples in the figures. For example, figure 2,3 and 5 from Han et al.’s manuscript (Stem Cell Res Ther, 2017 Sep 29;8(1):210. doi: 10.1186/s13287-017-0663-6) showed how the significant difference should be labeled when one-way ANOVA is used. Please also indicate in the method section which post hoc test was used for one-way ANOVA.
b. Please carefully interpret the data, as several statements in the results section are not true. For example, the statement in line 222-223 is not true. Figure 1 panels A and B clearly show that even at 12 and 24h, the TNF-alpha and IL-6 levels were significantly higher than the baseline, which indicate that the inflammatory state of the cells did not return to normal. The statement in line 252-253 is not true. Figure 2D shows that IL-6 protein level was increased in the inflammatory exosome group. The statement in line 255-256 is not true. Figure 2C shows that TNF-α protein level was decreased in the CM+GW4869 group.
2 Please indicate what “control” is in figure 1H. Why the extracted exosomes showed no CD81 expression?
3 I have a hard time understanding figure 1G. It would be help if the authors could provide some elaborations in the results section.
4 In figure 2A, it would be better if the authors could provide pictures of the staining of BMSCs before they were treated with exosomes as negative control.
5 In the abstract, the conclusion the authors drew in line 53 is not supported by the evidence the authors provided. Bone tissue destruction is mainly caused by osteoclast-mediated bone resorption. It would be more proper to say “a promising method for bone tissue regeneration in inflammatory states”.

Additional comments

1 The discussion section should provide in depth discussion of the results. However, the current discussion was written in a way that is more like an introduction. Many literatures were cited on the purpose to provide broad and general information. In my opinion, more relevant literatures that show similar or contradictory results to the current study should be cited and discussed. In additional, the discussion should be concise. There is no need to repeat the content that is already involved in the results section.
2 I don’t think the statement in line 268 is true. It is the activity of osteoclasts that matters.
3 Please have the manuscript thoroughly checked by an editor. As currently written, the manuscript contains numerous grammar mistakes and awkwardly structured sentences.

·

Basic reporting

The manuscript meets PeerJ's publication standards and the content is relevant to the the topics published by PeerJ.
The authors provided sufficient background to highlight the importance of this work and help the audience understand the key context of this study

Experimental design

The authors did a great job at documenting all the methods and materials used for this study. Experimental designs were clearly laid out. The key research question (exploring the role of macrophage-derived exosome on osteogenic differentiation of BMSC) is clearly defined and will fill the knowledge gap with the findings from this study.

I only have a few minor comments on the appropriate control:
1. for Figure 2A: it is difficult to tell whether the PHK67-labeled exosomes are actually concentrated around the nuclei of BMSCs or just randomly distributed in the cytoplasm; providing additional background staining as control would help clarify this (e.g. F-actin staining to show the cell body of BMSCs, ER staining to show the area adjacent to the nuclei); a negative control with just the PKH67 antibody and the plain medium / solvent used to suspend the exosomes should also be provided

2. for Figure 2B-2F: a baseline control of TNFa and IL-6 levels in untreated BMSCs is needed (i.e., BMSCs treated with the plain medium / solvent used to suspend the exosomes, and BMSCs treated with the plain medium used to make conditional medium); adding such baseline control will provide more context for the change in TNFa and IL-6 levels because it looks like just adding the ordinary exosomes also increased TNFa and IL-6 levels

3. Labeling in Figure 2B-2F: Com-exo is not clearly documented / explained in the rest of the manuscript, also not a straightforward abbreviation for ordinary exosome. It would be great if the authors can add the appropriate labeling during the revision process.

Validity of the findings

This study have provided some very interesting findings on what roles macrophages play in the process of BMSC osteogenic differentiation. It's clear that the exosomes secreted by macrophages after inflammatory response suppress osteogenic gene expression in BMSCs.

The only unclear link is whether this suppression is caused by the inflammatory response: i.e., inflammatory exosomes led to increased inflammatory gene expression in BMSCs, which led to decreased ALP, Col1, and Runx2 expression; or it is something else in the inflammatory macrophage that led to the increase of osteogenic expression. The current conclusion points to the former but the author did not provide sufficient evidence to validate this causal relationship.

To fully validate this, the authors should consider treating the BMSCs with just TNFa or IL-6, to see if that alone suppressed the osteogenic gene expression.

Additional comments

No comment. Great job on this study!

---

## Round 0.2 · accepted · Accept

The authors appropriately answered all the comments.

·

Basic reporting

Authors have successfully responded to my comments.

Experimental design

None.

Validity of the findings

None.

Additional comments

None.

Reviewer 2 ·

Basic reporting

Authors appropriately answered all the comments.

Experimental design

Authors appropriately answered all the comments.

Validity of the findings

Authors appropriately answered all the comments.

Additional comments

Authors appropriately answered all the comments.